# Heart rate and diastolic arterial pressure in cardiac arrest patients: A nationwide, multicenter prospective registry

Chul Han[1], Jae Hoon Lee [2]*, on behalf of the Korean Hypothermia Network Investigators[¶]

1 Department of Emergency Medicine, Seoul Hospital, EWHA Womans University, Seoul, South Korea,
2 Department of Emergency Medicine, Dong-A University College of Medicine, Busan, South Korea

¶ Membership of the Korean Hypothermia Network is provided in the Acknowledgments.
* leetoloc@dau.ac.kr

**Data Availability Statement:** All relevant data are within the manuscript.

**Funding:** This work was supported by the Dong-A University Research Fund (www.donga.ac.kr). The

## Abstract

### Background

Guidelines have recommended monitoring mean arterial pressure (MAP) and systolic arterial pressure (SAP) in cardiac arrest patients, but there has been relatively little regard for diastolic arterial pressure (DAP) and heart rate (HR). We aimed to determine the prognostic significance of hemodynamic parameters at all time points during targeted temperature management (TTM).

### Methods

We reviewed the SAP, DAP, MAP, and HR data in out-of-hospital cardiac arrest (OHCA) survivors from the prospective multicenter registry of 22 teaching hospitals. This study included 1371 patients who underwent TTM among 10,258 cardiac arrest patients. The hemodynamic parameters were recorded every 6 hours from the return of spontaneous circulation (ROSC) to 4 days. The risks of those according to time points during TTM were compared.

### Results

Of the included patients, 943 (68.8%) had poor neurological outcomes. The predictive ability of DAP surpassed that of SAP and MAP at all time points, and among the hemodynamic variables HR/DAP was the best predictor of the poor outcome. The risks in patients with DAP < 55 to 70 mmHg and HR > 70 to 100 beats/min were steeply increased for 2 days after ROSC and correlated with the poor outcome at all time points. Bradycardia showed lower risks only at 6 hours to 24 hours after ROSC.

### Conclusion

Hemodynamic parameters should be intensively monitored especially for 2 days after ROSC because cardiac arrest patients may be vulnerable to hemodynamic instability during TTM. Monitoring HR/DAP can help access the risks in cardiac arrest patients.

funders had no role in study design, data collection and analysis, decision to publish, or preparation of the manuscript.

**Competing interests:** The authors have declared that no competing interests exist.

## Introduction

Hemodynamic monitoring is essential in cardiac arrest patients and hemodynamic parameters such as systolic arterial pressure (SAP) and mean arterial pressure (MAP) have been primarily used for hemodynamic monitoring as recommended in the most recent guidelines for cardiac arrest patients [1]. Furthermore, most studies on hypotension episodes in cardiac arrest patients have been studied with MAP or SAP for neuroprognostication [2]. However, two studies demonstrated that DAP in the early phase of admission can be superior to SAP or MAP for neuroprognostication in cardiac arrest patients [3, 4]. Unfortunately, the evidence of these studies remains weak because they were not multi-center large-scale studies and had small populations. Moreover, the authors only investigated the hemodynamic data within 6 hours.

Diastolic arterial pressure (DAP) is being proposed as a promising prognostic tool in addition to SAP or MAP. DAP may be as available as SAP to assess the prognosis or risks in septic shock that can be mixed with hypovolemic, cardiogenic, and distributive shock [5, 6]. Another study revealed that DAP is superior to SAP in evaluating the prognosis of cardiogenic shock [7]. The role of DAP for risk assessment in cardiac arrest patients requires new consideration as hypotension during cardiac arrest can result from a number of different mechanisms.

Additionally, HR is another hemodynamic variable that could serve as a risk factor in cardiac arrest patients [8]. Hypotension is compensated for by increased sympathetic activity during hemorrhagic shock and acute critical illness [9, 10], and eventually HR increases in accordance with increased sympathetic activity. As a reflection of this compensatory mechanism, the shock index (HR/SAP), modified shock index (HR/MAP), and diastolic shock index (HR/DAP) have been used to predict prognosis in various diseases [5, 11]; however, the diastolic shock index has never been compared with other hemodynamic parameters in cardiac arrest patients. Also, the significance of DAPs during an entire time window of targeted temperature management (TTM) has never been debated.

The primary aim of this study was to explore the neuroprognostic significance of certain hemodynamic variables such as HR and DAP in comparison with SAP and MAP and their respective shock indices in survivors of out-of-hospital cardiac arrest (OHCA) during the post- return of spontaneous circulation (ROSC) phase.

## Materials and methods

### Study design and setting

This prospectively conducted multicenter observational cohort study was based on the Korean Hypothermia Network prospective registry (KORHN-PRO). The KORHN is a multicenter clinical research consortium for TTM in South Korea. Among 32 hospitals, 22 teaching hospitals throughout South Korea participated in this study and collected data from OHCA patients treated with TTM in advanced critical care settings. This study was approved by the institutional review boards of all participating hospitals and registered at the International Clinical Trials Registry Platform (NCT02827422). The Dong-A University Hospital Institutional Review Board (IRB) approved the study under entry code DAUHIRB-16-079. Written informed consent was obtained from all patients' legal surrogates. The data were regularly monitored and reviewed by three clinical research associates, the investigator, and the clinical research coordinator of each site, with feedback from the investigator of the corresponding site. The 22 centers used a standardized TTM protocol across all sites, however, administration of vasoactive drugs and fluids and rewarming time were managed in accordance with institutional practices.

## Study population

Among 10,258 cardiac arrest patients enrolled between October 2015 and December 2018, 1371 comatose survival patients were included. The inclusion criteria were as follows: patients over 18 years old, patients treated with mild therapeutic hypothermia after OHCA, and patients with an unconscious mental status (Glasgow Coma Scale < 8) after ROSC. Patients were excluded from the study who had rearrest events or death within 24 hours on admission (because TTM setting may be changed and missing data were increased), acute stroke (because TTM for 7 days was performed), a do not resuscitate (DNR) order, a prearrest cerebral performance category (CPC) score of 3 or 4, disease that would make survival at 6 months unlikely, a body temperature of <30˚C on admission, and patients whose caregiver did not sign the written informed consent form.

## Data collection

Blood pressure and heart rate were investigated every 6 hours for 4 days after ROSC via an arterial line or a noninvasive blood pressure cuff. SAP and DAP were examined and MAP was calculated from SAP and DAP. The time duration, maintenance dose, and total dose of vasopressors and inotropes including norepinephrine, vasopressin, dopamine, epinephrine, and dobutamine were recorded. The data on SAP, DAP, MAP, HR, and vasoactive agents were gathered from nurses' records and chart reviews.

To consider the various confounding factors that are related to poor neurological outcome, basal demographic, resuscitation, and post-resuscitation variables were obtained from the prospective registry. Furthermore, various therapeutic procedures, such as percutaneous coronary intervention, renal replacement therapy, extracorporeal membrane oxygenator, and external cardiopulmonary resuscitation (E-CPR) that can influence neuroprognostication were additionally reviewed. The primary outcome was CPC scores, which were investigated by progress notes in each hospital or direct phone call to transferred hospitals or caregivers after 6 months; a CPC of 3 to 5 was considered a poor neurological outcome. Other risk scores for neuroprognostication, such as the Four score and cardiovascular sequential organ failure assessment (SOFA) score, were extracted and compared with the hemodynamic variables.

## Hemodynamic variables

All the hemodynamic variables, including SAP, DAP, MAP, and HR, were compared for neuroprognostication in fixed time points (6 hour interval for 4 days), and the best combination was extrapolated from the relationship. We scrutinized the changes in blood pressure and HR according to time point. The hemodaynamic variables were measured for 4 days after ROSC, but cases who died one day after admission were regarded as missing hemodynamic variable data and the missing data were excluded in the analysis.

## Statistical analysis

Descriptive statistics included percentages to summarize categorical variables and medians and interquartile ranges to summarize continuous variables. Inferential statistics included Fisher's exact test to compare categorical variables and the Mann-Whitney U test to compare continuous variables. The risks associated with the level of DAP were assessed as odds ratios (ORs) according to time points using cubic spline models with 95% confidence intervals. Moreover, the accuracies of DAP and HR for neuroprognostication were compared over time. After the multivariable logistic regression analysis allowed for significant covariates, the adjusted OR regarding HR/DAP was analyzed. The discriminative ability of the hemodynamic variables for neuroprognostication was evaluated using receiver operating characteristic

(ROC) curves with the corresponding area under the curve (AUC) and related sensitivity, specificity, and positive and negative predictive value.

## Results

### Participation and basal characteristics

Of 1371 patients assessed for eligibility, 943 patients (68.8%) had poor neurological outcomes while 428 patients (31.2%) had good neurologic outcomes and 794 patients (57.9%) died while 577 patients (42.1%) survived. Multivariable analysis was performed to test the multiplicative interaction in the association between the hemodynamic variables and a number of variables, such as basal demographic, resuscitation, and post-resuscitation variables. Significant variables included age, male sex, witnessed arrest, bystander CPR, low flow time, no flow time, prehospital ECG rhythm, causes of cardiac arrest, prearrest CPC, pupil reflex, GCS motor function, Four score, cardiovascular SOFA score, fluid balance, lactate, creatinine, and total dose of vasoactive agents (Table 1). The group with poor outcomes undoubtedly had poor neurologic examinations, more use of vasopressors, and more fluid administration. With allowance made for the statistically significant covariates, namely, age, sex, witnessed arrest, low flow time, shockable rhythm, pupil reflex, GCS motor, and lactate level, the risk of HR/DAP regarding poor outcome was superior to the cardiovascular SOFA score (adjusted OR 1.7 vs 1.279, Table 2).

### Hemodynamic parameters

Blood pressures and HRs at all time points for 4 days after ROSC were measured, and the predictive powers of these parameters are compared in Table 3. The predictive ability of DAP surpassed that of SAP and MAP at all time points, and neither SAP nor MAP were greater predictors than DAP (Table 3). The AUC and cut-off value of DAP at 0, 6, 12, 18, 24, 30, 36, 42, and 48 hours after ROSC were as follows: 0.632 and 71 mmHg; 0.598 and 65; 0.567 and 73; 0.557 and 56; 0.564 and 67; 0.573 and 65; 0.59 and 60; 0.585 and 62; and 0.585 and 62, respectively, and those of HR at 0, 6, 12, 18, 24, 30, 36, 42, and 48 hours after ROSC were as follows: 0.544 and 111 beats/min; 0.605 and 100; 0.662 and 73; 0.684 and 79; 0.656 and 72; 0.639 and 75; 0.65 and 83; 0.62 and 103; and 0.582 and 95, respectively. Compared with SAP, DAP, MAP, and HR, HR/DAP among all combinations of hemodynamic variables was the best predictor of neuroprognostication at all time points.

Interestingly, the accuracies of DAP for neuroprognostication during TTM revealed an inverse U-shape, increasing and then decreasing, while those of HR showed a U-shape, decreasing and then increasing (Fig 1). DAP had lower prediction power of neurological outcome than HR for 2 days. Additionally, DAP < 55 to 70 mmHg for 2 days was associated with increased risks of poor neurologic outcome, but the risks of hypotension episodes were steeply increased up to 48 hours after ROSC and then gradually decreased 72 hours after ROSC (Fig 2). The risks of hypertension episodes were slightly exhibited over 72 hours after ROSC. Meanwhile, HR > 72–103 beats/min (cut-off) within 2 days after ROSC led to poor neurological outcomes, and the risk of HR > 70 to 100 beats/min was gradually restored after increasing for 2 days. Bradycardia or HR < 60 beats/min showed lower risks 6 hours to 24 hours after ROSC and higher risks at ROSC and more than 48 hours after ROSC (Fig 2). Bradycardia between 24 and 48 hours after ROSC had no effect on risk.

## Discussion

The predictive ability of DAP for risk assessment was superior to that of SAP or MAP in cardiac arrest patients and HR/DAP or diastolic shock index among all hemodynamic variables

**Table 1. Baseline characteristics.**

| | All patients (n = 1371) | Good outcome (n = 428) | Poor outcome (n = 943) | p value[a] |
|---|---|---|---|---|
| **Age, years** | 62 (51–74) | 58 (48–66) | 65 (53–77) | <0.001 |
| **Male, n (%)** | 975 (71.1) | 333 (77.8) | 642 (68.1) | <0.001 |
| **BMI, kg/m²** | 23.3 (20.9–25.7) | 23.3 (21.3–25.6) | 23.4 (20.8–25.7) | 0.823 |
| **Witnessed arrest, n (%)** | 949 (70) | 361 (84.5) | 588 (63.3) | <0.001 |
| **Bystander CPR, n (%)** | 843 (62.4) | 292 (69.2) | 551 (59.3) | 0.001 |
| **Time from arrest to CPR start, minutes** | 1 (0–7) | 1 (0–5) | 1 (0–8) | 0.005 |
| **Time from CPR start to ROSC, minutes** | 15 (9–22.75) | 15 (9–22.8) | 31 (20–42) | <0.001 |
| **Time from ROSC to TTM start, hours** | 3.4 (2.2–4.9) | 3.6 (2.5–5) | 3.3 (2–4.8) | 0.002 |
| **Prehospital ECG rhythm** | | | | <0.001 |
| **Asystole, n (%)** | 445 (37) | 23 (6.1) | 422 (51) | |
| **PEA, n (%)** | 269 (22.3) | 54 (14.3) | 215 (26) | |
| **Pulseless VT, n (%)** | 15 (1.2) | 11 (2.9) | 4 (0.5) | |
| **VF, n (%)** | 448 (37.2) | 272 (72.1) | 176 (21.3) | |
| **ROSC, n (%)** | 27 (2.2) | 17 (4.5) | 10 (1.2) | |
| **Previous history** | | | | |
| **Cardiovascular disease[b], n (%)** | 285 (20.8) | 99 (34.7) | 186 (19.7) | 0.152 |
| **Neurologic disease[c], n (%)** | 138 (10.1) | 27 (6.3) | 111 (11.8) | 0.002 |
| **Pulmonary disease, n (%)** | 106 (7.7) | 13 (3) | 93 (9.9) | <0.001 |
| **Malignancy, n (%)** | 80 (5.8) | 23 (5.4) | 57 (6) | 0.465 |
| **Psychologic disease, n (%)** | 51 (3.7) | 5 (1.2) | 46 (4.9) | <0.001 |
| **Causes of cardiac arrest** | | | | <0.001 |
| **Medical, n (%)** | 851 (62.1) | 479 (50.8) | 372 (86.9) | |
| **Trauma, n (%)** | 28 (2) | 2 (0.5) | 26 (2.8) | |
| **Submersion, n (%)** | 19 (1.4) | 4 (0.9) | 2 (0.2) | |
| **Electrocution, n (%)** | 6 (0.4) | 3 (0.5) | 3 (0.4) | |
| **Drug overdose, n (%)** | 16 (1.2) | 5 (1.2) | 11 (1.2) | |
| **Asphyxia, n (%)** | 78 (5.7) | 6 (1.4) | 72 (7.6) | |
| **Hanging, n (%)** | 160 (11.7) | 12 (2.8) | 148 (15.7) | |
| **Others, n (%)** | 213 (15.5) | 23 (5.4) | 190 (20.1) | |
| **Pre-arrest CPC** | 1 (1–1) | 1 (1–1) | 1 (1–1) | <0.001 |
| **Pupil reflex, n (%)** | 643 (47.3) | 346 (81) | 297 (31.8) | <0.001 |
| **GCS motor, score** | 1 (1–1) | 1 (1–3) | 1 (1–1) | <0.001 |
| **Four score[d]** | 0 (0–3) | 4 (0–7) | 0 (0–1) | <0.001 |
| **Cardiovascular SOFA[e] at day 1** | 4 (2–4) | 3 (0–4) | 4 (3–4) | <0.001 |
| **Total dose of dopamine, μg** | 5605 (1650–21600) | 2630 (1200–6060) | 7800 (1882–22903) | <0.001 |
| **Total dose of norepinephrine, μg** | 108 (30–360) | 46 (18–136.2) | 150 (38.4–480) | <0.001 |
| **Total dose of vasopressin, IU** | 31 (7.2–113.4) | 47 (16.4–291.2) | 30.6 (7.2–108) | 0.303 |
| **Total dose of epinephrine, μg** | 67.8 (18.3–270.8) | 46.8 (9.9–153) | 72 (22.2–294) | 0.24 |
| **Total dose of dobutamine, μg** | 3600 (800–15970) | 3360 (804–12908) | 4200 (800–16709) | 0.689 |
| **Input/Output at day 1, Ml** | 450 (-354–1725) | -114 (-744.8–756.3) | 792 (-111–2117.8) | <0.001 |
| **Initial lactate, mg/dL** | 9.7 (6.1–12.9) | 7.1 (4.3–10.9) | 10.6 (7.5–13.6) | <0.001 |
| **Initial creatinine, mg/dL** | 1.3 (1.1–1.8) | 1.2 (1–1.4) | 1.4 (1.1–2.2) | <0.001 |
| **Target temperature, ˚C** | 33 (33–34) | 33 (33–34) | 33 (33–34) | 0.445 |
| **Duration of TTM, hours** | 24 (24–24) | 24 (24–24) | 24 (24–24) | 0.022 |
| **PCI, n (%)** | 206 (41.2) | 117 (40.3) | 89 (42.4) | 0.713 |
| **RRT, n (%)** | 249 (18.3) | 31 (7.3) | 218 (23.3) | <0.001 |
| **ECMO, n (%)** | 48 (3.5) | 21 (4.9) | 27 (2.9) | 0.08 |

*(Continued)*

**Table 1.** (Continued)

|  | All patients (n = 1371) | Good outcome (n = 428) | Poor outcome (n = 943) | p value[a] |
|---|---|---|---|---|
| ECPR, n (%) | 15 (1.1) | 2 (0.5) | 13 (1.4) | 0.167 |

Values are expressed as number (%) or median (interquartile range).

BMI denotes body mass index; CPR, cardiopulmonary resuscitation; ROSC, restoration of spontaneous circulation; TTM, targeted temperature management; PEA, pulseless electric activity; VT, ventricular tachycardia; PCI, percutaneous coronary intervention; RRT, renal replacement therapy; ECMO, extra-corporeal membrane oxygenation; ECPR, external cardiopulmonary resuscitation.

[a] The p value was calculated by means of Fisher's exact test and the Mann-Whitney U-test.

[b] Cardiovascular disease included diseases such as cardiac arrest, coronary artery disease, and congestive heart failure.

[c] Neurological disease included diseases such as transient ischemic accident, stroke, and other neurological diseases.

[d] The four-scale score consisted of eye response, motor response, brainstem reflexes, and respiration and ranged from 0 to 4.

[e] Scores on the cardiovascular SOFA ranged from 0 to 4 (0, no hypotension; 1, MAP <70 mmHg; 2, dopamine $\leq$ 5 µg/kg/min or dobutamine; 3, dopamine > 5 µg/kg/min or epinephrine $\leq$ 0.1 µg/kg/min or norepinephrine $\leq$ 0.1 µg/kg/min; 4, dopamine > 15 µg/kg/min or epinephrine > 0.1 µg/kg/min or norepinephrine > 0.1 µg/kg/min).

was the best predictor of poor neurological outcomes at all time points. DAP < 55 to 70 mmHg and HR > 70 to 100 beats/min for 2 days after ROSC were correlated with poor neurological outcomes, HR < 60 beats/min 6 to 24 hours after ROSC showed a better outcome, and HR < 60 beats/min 48 hours after ROSC revealed a worse outcome.

DAP reflects vascular tone and arterial compliance [12]. This vascular tone or systemic vascular resistance (SVR) in cardiac arrest patients can be affected by post-resuscitation syndrome, including cardiac stunning and vasodilation, which may be maintained up to 72 hours [13]. "Sepsis"-like syndrome, which is also characterized by a systemic ischemic/reperfusion response, can occur in not only post-resuscitation syndrome or septic shock but also in cardiogenic shock [14, 15]. Simple hypovolemia might be more connected with SAP than DAP [16], however, diverse diseases or situations from sepsis, coronary ischemia, arrhythmia, or using vasopressor agents can combine in most cardiac arrest patients, and low blood pressure results from a certain combination of hypovolemic, cardiogenic, and distributed shock rather than pure shock. These multiplicative hemodynamics may influence DAP, reflecting vascular tone rather than SAP. The fact that the predictive ability of DAP may surpass that of SAP in reflecting hemodynamic status shows the probability that the diastolic shock index may be as available as the shock index or modified shock index.

The reason that DAP is a crucial factor in the heart and brain in cardiac arrest patients is as follows. DAP is a major determinant of coronary perfusion pressure, which is essential to

**Table 2. Multi-variable analysis to predict poor neurological outcome.**

| Standardized variables | Odds ratio | *p* value | 95% CI |
|---|---|---|---|
| Age | 1.911 | <0.001 | 1.524–2.396 |
| Male | 0.783 | 0.022 | 0.635–0.965 |
| Witnessed arrest | 0.665 | 0.001 | 0.527–0.84 |
| Time from CPR start to ROSC | 3.296 | <0.001 | 2.453–4.429 |
| Shockable rhythm | 0.403 | <0.001 | 0.331–0.491 |
| Pupil reflex | 0.64 | <0.001 | 0.511–0.8 |
| GCS motor | 0.673 | <0.001 | 0.551–0.821 |
| Four score | 0.589 | <0.001 | 0.47–0.738 |
| Initial lactate | 1.37 | 0.009 | 1.083–1.733 |
| Cardiovascular SOFA | 1.279 | 0.012 | 1.057–1.548 |
| HR/DAP | 1.7 | <0.001 | 1.319–2.192 |

**Table 3. Predictive accuracy of blood pressure and heart rate according to timing.**

|  | Outcome | Sensitivity | Specificity | PPV | NPV | AUC |
|---|---|---|---|---|---|---|
| SAP in ROSC | Good | 38.6 | 76.5 | 78.5 | 35.9 | 0.59 |
| DAP in ROSC | Good | 56 | 65.77 | 78.4 | 40.2 | 0.632 |
| MAP in ROSC | Good | 57.5 | 61.4 | 76.8 | 39.4 | 0.617 |
| HR in ROSC | Poor | 47.2 | 65.5 | 75.1 | 36 | 0.544 |
| HR/DAP in ROSC | Poor | 39.1 | 82.3 | 82.9 | 38 | 0.636 |
| SAP in 6 hours | Good | 44.8 | 70.4 | 76.5 | 37.2 | 0.588 |
| DAP in 6 hours | Good | 40.3 | 74.9 | 77.6 | 36.9 | 0.598 |
| MAP in 6 hours | Good | 46.6 | 70.9 | 77.5 | 38.2 | 0.601 |
| HR in 6 hours | Poor | 51.6 | 72.6 | 80.1 | 41.1 | 0.65 |
| HR/DAP in 6 hours | Poor | 51.6 | 72.6 | 80.1 | 41.1 | 0.65 |
| SAP in 12 hours | Good | 31.4 | 78.8 | 75.3 | 35.8 | 0.545 |
| DAP in 12 hours | Good | 40.3 | 73.8 | 76 | 37.5 | 0.567 |
| MAP in 12 hours | Good | 26.7 | 86.1 | 79.8 | 36.3 | 0.562 |
| HR in 12 hours | Poor | 71 | 54.2 | 76.2 | 47.5 | 0.662 |
| HR/DAP in 12 hours | Poor | 52.9 | 75.2 | 81.4 | 43.7 | 0.679 |
| SAP in 24 hours | Good | 19.1 | 91.4 | 81.8 | 36 | 0.542 |
| DAP in 24 hours | Good | 46.1 | 64.8 | 72.5 | 37.4 | 0.564 |
| MAP in 24 hours | Good | 32.9 | 80 | 76.8 | 37.2 | 0.559 |
| HR in 24 hours | Poor | 65.1 | 60.7 | 77 | 46.2 | 0.656 |
| HR/DAP in 24 hours | Poor | 47.1 | 79.3 | 82.1 | 42.8 | 0.666 |
| SAP in 48 hours | Good | 32.1 | 82 | 76.2 | 40.3 | 0.588 |
| DAP in 48 hours | Good | 54.4 | 65.7 | 73.9 | 44.6 | 0.628 |
| MAP in 48 hours | Good | 38.7 | 78 | 76 | 41.5 | 0.594 |
| HR in 48 hours | Poor | 49.7 | 65.6 | 72.1 | 42.1 | 0.582 |
| HR/DAP in 48 hours | Poor | 54.4 | 65.7 | 73.9 | 44.6 | 0.628 |
| SAP in 72 hours | Good | 46.2 | 68.2 | 70.6 | 43.5 | 0.569 |
| DAP in 72 hours | Good | 56.1 | 60 | 69.8 | 45.3 | 0.593 |
| MAP in 72 hours | Good | 45.4 | 70.9 | 72 | 44 | 0.587 |
| HR in 72 hours | Poor | 34.5 | 76 | 70.4 | 41.3 | 0.538 |
| HR/DAP in 72 hours | Poor | 52.7 | 62.5 | 69.9 | 44.5 | 0.591 |
| SAP in 96 hours | Good | 39.4 | 69 | 66.2 | 42.5 | 0.54 |
| DAP in 96 hours | Good | 69.1 | 45.3 | 66 | 48.8 | 0.576 |
| MAP in 96 hours | Good | 61.2 | 50.9 | 65.7 | 46 | 0.566 |
| HR in 96 hours | Poor | 33.1 | 74.7 | 67 | 41.9 | 0.526 |
| HR/DAP in 96 hours | Poor | 47.5 | 68.4 | 69.9 | 45.7 | 0.581 |

PPV denotes positive predictive value; NPV, negative predictive value; SAP, systolic arterial pressure; DAP, diastolic arterial pressure; MAP, mean arterial pressure: HR, heart rate.

retain adequate myocardial perfusion in the post-resuscitation phase [17]. In addition, a lower threshold of cerebral autoregulation is often shifted rightward in cardiac arrest patients. An MAP of 70 mmHg (MAP reflects twice as much DAP as SAP) may still result in brain hypo-perfusion, even though it is within the normal range [18]. Therefore, maintaining a high normal DAP may help heart and brain resuscitation.

The predictive ability of DAP within 6 hours in cardiac arrest patients has been proven. Compared with SAP, MAP, and cardiovascular SOFA, DAP was a powerful predictor of poor neurological outcomes [3, 4]. In our results, the risk of poor outcome due to low DAP consistently soared for 2 days after ROSC and afterwards subsequently tapered off, but it still

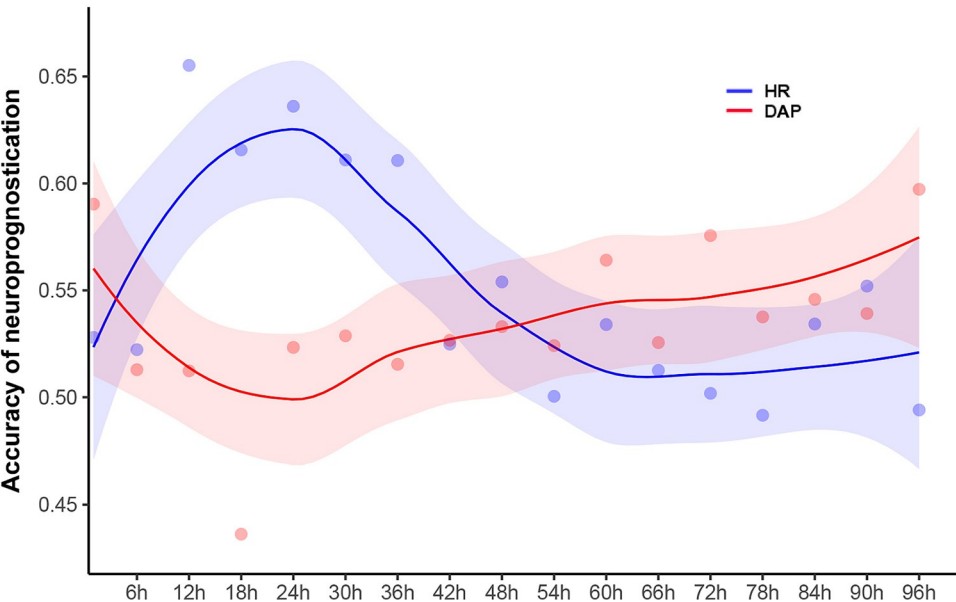

**Fig 1. Accuracy of heart rate and diastolic arterial pressure changes to predict poor outcome according to time points.**

sustained a certain level of risk. Similar results are reported for septic shock. The ability of DAP to predict poor outcome in septic shock patients was superior to that of SAP and showed a similar performance to the SOFA score [5]. Higher DAP was a powerful predictor of in-hospital survival, compared with SAP [6]. Additionally, high DAP 72 hours after ROSC indicated a slight risk of poor outcome in our study. Chronic adaptation to hypertension might change the risks, as patients with a history of hypertension in hospital cardiac arrest showed good outcomes with high MAP, while patients without a history of hypertension showed poor outcomes with MAP > 115 mmHg [19].

Myocardial stunning can often induce low stroke volume and systolic and diastolic dysfunction, but reduced HR and increased SVR dominate the clinical hemodynamic effects of TTM [20]. Patients who underwent TTM maintained a lower HR and higher SVR during TTM than those who did not undergo TTM without a significant difference in MAP or stroke volume (SV) [21, 22]. Although a low HR may persist during TTM, acute onset tachycardia in this setting portends a worse outcome. In intensive care unit (ICU) patients, but not cardiac arrest patients, new onset prolonged sinus tachycardia as a consequence of sympathetic activity has been associated with increased major cardiovascular events and higher mortality rates [23]. Similar to ICU patients with tachycardia, but with a lower HR of over 72 to 83 beats/min due to functional hypothermia downregulation, cardiac arrest patients were related to a poor outcome, especially 2 days after ROSC. Studies in cardiac arrest patients demonstrated that a higher HR is connected with a poor outcome [8], and lower HR indicates a good outcome [21, 24, 25]. Cardiac arrest patients treated with TTM with a heart rate <60 beats/min or sinus bradycardia <50 beats/min have shown good outcomes 8 hours after ROSC or at any time during hypothermia [24, 25]; however, these results were only measured during the early phase of TTM. Bradycardia at 6 hours to 24 hours showed lower risks of a poor outcome in our data, but the risks gradually increased over 48 hours after ROSC.

The shock index (HR/SAP) has been associated with clinical outcomes in sepsis, septic shock, trauma, and cardiovascular disease [11]. Compared with the predictive ability of the

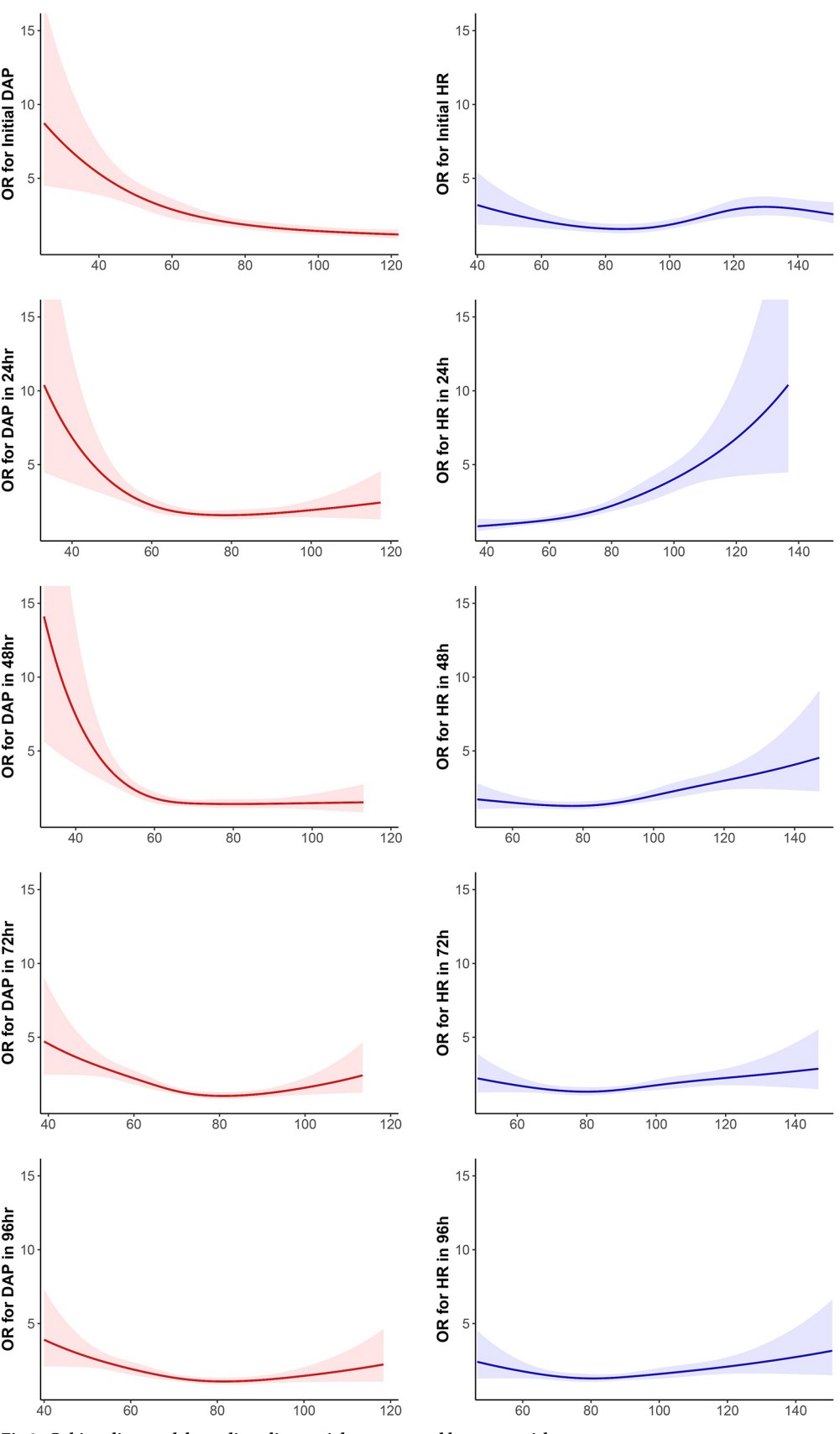

**Fig 2. Cubic spline models on diastolic arterial pressure and heart rate risks.**

shock index, that of the modified shock index is a better predictor of clinical outcome in most diseases other than hypovolemic shock [26, 27]. Even the modified shock index was similar or slightly superior to the shock index in predicting clinical outcome in trauma patients with hypovolemia [11, 28, 29]. The DAP or diastolic shock index in diseases with a complex interaction between various shock patterns would better reflect vasodilation than SAP or MAP [5]. Especially in septic shock patients, it has been demonstrated that the diastolic shock index is more effective in predicting clinical outcomes than the shock index [5]. Additionally, low DAP and high HR in patients with chronic AR are intimately connected with all causes of death, and the spline models of DAP and HR in the study correspond to our data [30]. The diastolic shock index in cardiac arrest has yet to be reported, and our results indicate that the diastolic shock index in cardiac arrest patients may have a pivotal role for risk assessment.

## Limitations

Our study has several limitations. First, despite using the prospective multicenter registry of most variables, hemodynamic variables were recorded via chart records. This retrospective nature might be a limitation. Second, the group that did not undergo TTM was not included. The DAP and HR in cardiac arrest patients who did not undergo TTM that can induce lower HR and higher SVR can show different hemodynamic features. The pure changes and characteristics in DAP and HR due to cardiac arrest in the beginning must be studied. Third, the patients who survived 24 hours after ROSC were included in our study, but the hemodynamic data were omitted in the analysis: 87 patients who died within the first 24 hours and 265 patients who were missed according to time points. There was not much missing data. Finally, HR/DAP had low accuracy in predicting poor outcomes in cardiac arrest patients. Nonetheless, HR/DAP would be available for risk stratification and assessment or as a risk factor to predict poor neurological outcomes along with other significant variables.

## Conclusions

Because TTM induces low HR and high SVR, increased HR and decreased DAP during TTM can be strongly linked with severe heart and brain damage. HR/DAP as a risk factor for poor neurological outcome in cardiac arrest patients was superior to SAP, MAP, HR, and cardiovascular SOFA score. DAP < 55 to 70 mmHg and HR > 70 to 100 beats/min for 2 days after ROSC were correlated with poor neurological outcome. Low DAP and high HR should be intensively monitored, especially for 2 days after ROSC, because hemodynamic changes at an early phase were associated with poor neurological outcome. Monitoring HR/DAP can help physicians guide the risk management of poor neurological outcomes in cardiac arrest patients.

## Acknowledgments

The following investigators participated in the Korean Hypothermia Network. Chair: Kyung Woon Jeung (Chonnam National University Hospital, E-mail: neoneti@hanmail.net). Principal investigators of each hospital: Kyu Nam Park (The Catholic University of Korea, Seoul St. Mary's Hospital); Minjung Kathy Chae (Ajou University Medical Center); Won Young Kim (Asan Medical Center); Byung Kook Lee (Chonnam National University Hospital); Dong Hoon Lee (Chung-Ang University Hospital); Tae Chang Jang (Daegu Catholic University Medical Center); Jae Hoon Lee (Dong-A University Hospital); Chul Han (Ewha womans university Seoul hospital); Yoon Hee Choi (Ewha Womans University Mokdong Hospital); Je Sung You (Gangnam Severance Hospital); Young Hwan Lee (Hallym University Sacred Heart Hospital); In Soo Cho (Hanil General Hospital); Su Jin Kim (Korea University Anam

Hospital); Jong-Seok Lee (Kyung Hee University Medical Center); Yong Hwan Kim (Samsung Changwon Hospital); Min Seob Sim (Samsung Medical Center); Jonghwan Shin (Seoul Metropolitan Government Seoul National University Boramae Medical Center); Yoo Seok Park (Severance Hospital); Hyung Jun Moon (Soonchunhyang University Hospital Cheonan); Won Jung Jeong (The Catholic University of Korea, St. Vincent's Hospital); Joo Suk Oh (The Catholic University of Korea, Uijeongbu St. Mary's Hospital); Seung Pill Choi (The Catholic University of Korea, Yeouido St. Mary's Hospital); Kyoung-Chul Cha (Wonju Severance Christian Hospital).

## Author Contributions

**Conceptualization:** Jae Hoon Lee.

**Data curation:** Chul Han, Jae Hoon Lee.

**Formal analysis:** Chul Han, Jae Hoon Lee.

**Investigation:** Chul Han, Jae Hoon Lee.

**Methodology:** Jae Hoon Lee.

**Project administration:** Chul Han, Jae Hoon Lee.

**Software:** Jae Hoon Lee.

**Supervision:** Jae Hoon Lee.

**Validation:** Chul Han, Jae Hoon Lee.

**Visualization:** Chul Han, Jae Hoon Lee.

**Writing – original draft:** Chul Han, Jae Hoon Lee.

**Writing – review & editing:** Chul Han, Jae Hoon Lee.

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
