## [Decision Letter · Decision Letter 0]

5 Jul 2022

PONE-D-22-07847

Heart Rate and Diastolic Arterial Pressure in Cardiac Arrest Patients: a Nationwide, Multicenter Prospective Registry

PLOS ONE

Dear Dr. Lee,

Thank you for submitting your manuscript to PLOS ONE. After careful consideration, we feel that it has merit but does not fully meet PLOS ONE’s publication criteria as it currently stands. Therefore, we invite you to submit a revised version of the manuscript that addresses the points raised during the review process.

We look forward to receiving your revised manuscript.

Kind regards,

Jignesh K. Patel

Academic Editor

PLOS ONE

https://journals.plos.org/plosone/s/file?id=ba62/PLOSOne_formatting_sample_title_authors_affiliations.pdf".

“This work was supported by the Dong-A University Research Fund.”

“The authors received no specific funding for this work.”

3. One of the noted authors is a group or consortium [The Korean Hypothermia Network Investigators]. In addition to naming the author group, please list the individual authors and affiliations within this group in the acknowledgments section of your manuscript. Please also indicate clearly a lead author for this group along with a contact email address.

Reviewers' comments:

Reviewer's Responses to Questions

**Comments to the Author**

1. Is the manuscript technically sound, and do the data support the conclusions?

Reviewer #1: Yes

2. Has the statistical analysis been performed appropriately and rigorously? 

Reviewer #1: I Don't Know

3. Have the authors made all data underlying the findings in their manuscript fully available?

Reviewer #1: Yes

4. Is the manuscript presented in an intelligible fashion and written in standard English?

Reviewer #1: Yes

5. Review Comments to the Author

Reviewer #1: This is an excellent review of shock indices and the role that DAP plays in neuroprognostication. I would include values for HR/DAP (along with HR/SAP, etc.) that offer the best prognostication, as this would make the study more applicable. I would also consider adding a piece about mortality in your text about HR/DAP.

6. PLOS authors have the option to publish the peer review history of their article (what does this mean?). If published, this will include your full peer review and any attached files.

Reviewer #1: No

---

## [Author Response · Author response to Decision Letter 0]

15 Jul 2022

Title of the Manuscript: Heart Rate and Diastolic Arterial Pressure in Cardiac Arrest Patients: a Nationwide, Multicenter Prospective Registry

Manuscript Number: PONE-D-22-07847

I appreciate detailed and helpful comments and recommendations of the reviewer on the submitted manuscript. Considering the comments and recommendations, I revised the manuscript as below. 

Reviewer: 1

This is an excellent review of shock indices and the role that DAP plays in neuroprognostication. I would include values for HR/DAP (along with HR/SAP, etc.) that offer the best prognostication, as this would make the study more applicable. 

▶ Thank you very much. As you commented, I performed multi-variable analysis with significant variables and made table 2 newly. This study aimed to compare hemodynamic variables in cardiac arrest patients. I thought it might not need to deal with the risks of other variables on poor neurological outcome here. However, multi-variable analysis seems to make the result showed objectively.

One thing, it could be a problem that the statistical comparison and analysis of HR/DAP and HR/SAP can cause multicollinearity including the same or similar variable and so it was not conducted for the reason. 

I would also consider adding a piece about mortality in your text about HR/DAP.

▶ I provided the information of survival rate in my text as below. 

“Of 1371 patients assessed for eligibility, 943 patients (68.8%) had poor neurological outcomes while 428 patients (31.2%) had good neurologic outcomes and 794 patients (57.9%) died while 577 patients (42.1%) survived.”

---

## [Editor Report · Decision Letter 1]

23 Aug 2022

Heart rate and diastolic arterial pressure in cardiac arrest patients: a nationwide, multicenter prospective registry

PONE-D-22-07847R1

Dear Dr. Lee,

We’re pleased to inform you that your manuscript has been judged scientifically suitable for publication and will be formally accepted for publication once it meets all outstanding technical requirements.

Kind regards,

Jignesh K. Patel

Academic Editor

PLOS ONE
---

## [Editor Report · Acceptance letter]

2 Sep 2022

PONE-D-22-07847R1 

Heart rate and diastolic arterial pressure in cardiac arrest patients: a nationwide, multicenter prospective registry 

Dear Dr. Lee:

I'm pleased to inform you that your manuscript has been deemed suitable for publication in PLOS ONE. Congratulations! Your manuscript is now with our production department. 

Kind regards, 

on behalf of

Dr. Jignesh K. Patel 

Academic Editor

PLOS ONE